# Resolvin D1 Decreases Severity of Streptozotocin-Induced Type 1 Diabetes Mellitus by Enhancing BDNF Levels, Reducing Oxidative Stress, and Suppressing Inflammation

**DOI:** 10.3390/ijms22041516

**Published:** 2021-02-03

**Authors:** Siresha Bathina, Undurti N. Das

**Affiliations:** 1BioScience Research Centre and Department of Medicine, Gayatri Vidya Parishad Hospital, GVP College of Engineering Campus, Visakhapatnam 530048, India; siresha99@googlemail.com; 2Department of Biotechnology, Gandhi Institute of Science (GIS), GITAM University, Visakhapatnam 530048, India; 3UND Life Sciences, 2221 NW 5th St, Battle Ground, WA 98604, USA; 4International Research Centre, Biotechnologies of the Third Millennium, ITMO University, 191002 Saint-Petersburg, Russia

**Keywords:** type 1 diabetes mellitus, streptozotocin, resolving D1, antioxidants

## Abstract

Type 1 diabetes mellitus is an autoimmune disease characterized by increased production of pro-inflammatory cytokines secreted by infiltrating macrophages and T cells that destroy pancreatic β cells in a free radical-dependent manner that causes decrease or absence of insulin secretion and consequent hyperglycemia. Hence, suppression of pro-inflammatory cytokines and oxidative stress may ameliorate or decrease the severity of diabetes mellitus. To investigate the effect and mechanism(s) of action of RVD1, an anti-inflammatory metabolite derived from docosahexaenoic acid (DHA), on STZ-induced type 1 DM in male Wistar rats, type 1 diabetes was induced by single intraperitoneal (i.p) streptozotocin (STZ-65 mg/kg) injection. RVD1 (60 ng/mL, given intraperitoneally) was administered from day 1 along with STZ for five consecutive days. Plasma glucose, IL-6, TNF-α, BDNF (brain-derived neurotrophic factor that has anti-diabetic actions), LXA4 (lipoxin A4), and RVD1 levels and BDNF concentrations in the pancreas, liver, and brain tissues were measured. Apoptotic (*Bcl2/Bax*), inflammatory (*COX-1/COX-2/Nf-κb/iNOS/PPAR-γ*) genes and downstream insulin signaling proteins (Gsk-3β/Foxo1) were measured in the pancreatic tissue along with concentrations of various antioxidants and lipid peroxides. RVD1 decreased severity of STZ-induced type 1 DM by restoring altered plasma levels of TNF-α, IL-6, and BDNF (*p* < 0.001); expression of pancreatic *COX-1/COX-2/PPAR-γ* genes and downstream insulin signaling proteins (Gsk-3β/Foxo1) and the concentrations of antioxidants and lipid peroxides to near normal. RVD1 treatment restored expression of *Bcl2/Pdx* genes, plasma LXA4 (*p* < 0.001) and RVD1 levels and increased brain, pancreatic, intestine, and liver BDNF levels to near normal. The results of the present study suggest that RVD1 can prevent STZ-induced type 1 diabetes by its anti-apoptotic, anti-inflammatory, and antioxidant actions and by activating the *Pdx* gene that is needed for pancreatic β cell proliferation.

## 1. Introduction

Type 1 diabetes (T1DM) is an autoimmune disease in which a subclass of T lymphocytes induces apoptosis of pancreatic β cells [1,2]. These patients need life-long insulin replacement therapy. Hitherto the focus has been on islet cell allografting [3], stem cell derived β cells [4], and immunotherapies using monoclonal antibodies targeting CD3 [5]. STZ-induced T1DM is a suitable model for studying β cell diabetic glucotoxicity as it partially destructs pancreas and reduces β cell mass [6]. STZ is eliminated within 48 h of administration and so its DNA methylating action diminishes quickly and cannot be detected after 24 h of exposure [6,7,8]. Thus, STZ acute toxicity is of short duration. Yet, these animals show continued deterioration of β cell function due to hyperglycemia established by acute STZ toxicity resulting in further β cell dysfunction [6,9,10]. It is noteworthy that once this hyperglycemia is reversed, β cell function improves via β cell regeneration [6,10,11]. Hence, employing low dose (45 mg/kg) instead of high dose (100 mg/kg) STZ is a suitable model to study not only T1DM but also ways and means of β cell regeneration. In addition, in such a STZ-induced T1DM model, measurement of *Pdx-1* gene expression would tell whether the interventions employed ameliorated or reduced the severity of hyperglycemia by virtue of its action on pancreatic β cell proliferation or not. The *Pdx-1* (pancreatic and duodenal homeobox 1) gene, also known as insulin promoter factor 1, is a transcription factor that is necessary for pancreatic development, including β-cell maturation and duodenal differentiation.

*Pdx1* is the earliest marker for pancreatic differentiation, with the fates of pancreatic cells controlled by downstream transcription factors. The initial pancreatic bud is composed of Pdx1+ pancreatic progenitor cells that proliferate and branch in response to FGF-10 signaling. The final stages of pancreas development involve the production of insulin-producing β-cells and glucagon-producing α-cells. *Pdx-1* is necessary for β-cell maturation. In the mature pancreas, *Pdx1* expression is required for the maintenance and survival of β-cells. Reduction in the level of *Pdx1* expression in the mature pancreas β-cells results in increased production of glucagon, suggesting that it (*Pdx-1*) inhibits the conversion of β-cells into α-cells. Furthermore, *Pdx-1* is needed to mediate the effect of insulin on the apoptosis of β-cells: a small concentration of insulin protects β-cells from apoptosis, but not in cells where *Pdx-1* expression has been inhibited [12,13,14,15].

In a previous study, we observed that ω-3 polyunsaturated fatty acids (PUFAs) and their anti-inflammatory metabolites resolvins and protectins and ω-6 arachidonic acid and its anti-inflammatory product lipoxin A4 (LXA4) protect rat pancreatic β cells (RIN5F cells) from the cytotoxic action of alloxan and streptozotocin (STZ) in vitro [16,17,18]. In addition, we showed previously that LXA4 can prevent alloxan-induced type 1 DM and streptozotocin-induced type 1 and type 2 diabetes mellitus in experimental animals [16,17,18]. Thus, ω-3 PUFAs: eicosapentaenoic acid (EPA), docosahexaenoic acid (DHA), and ω-6 arachidonic acid (AA) and their anti-inflammatory metabolites have immunomodulatory actions and preserve pancreatic β cell function. In an extension of these studies, we conducted the present in vivo study to evaluate potential anti-diabetic action of anti-inflammatory metabolite of DHA, resolvin D1. Resolvin D1 (RVD1) reduces inflammation and showed promise in chronic inflammatory diseases, such as rheumatoid arthritis, inflammatory bowel disease, and asthma [19,20,21,22,23,24]. However, its role in the prevention of T1DM has not been elucidated.

DHA is derived from dietary alpha-linolenic acid (ALA) by the action of enzymes desaturases and elongases [20]. DHA can also be obtained from diet since it is rich in marine fish. DHA is available as supplements over the counter in many countries. DHA and AA are particularly rich in human breast milk [25,26,27]. Studies revealed that breast fed children have low incidence of type 1 DM [28,29,30]. This coupled with the observation that cod liver oil (rich in EPA and DHA) taken during pregnancy was associated with reduced risk of type I diabetes in the offspring [31] led to the suggestion that possibly, AA, EPA, and DHA supplementation during infancy and early childhood may be responsible for this beneficial action. This is supported by the fact that human breast milk is rich in AA, EPA, and DHA and their anti-inflammatory metabolites such as lipoxin A4 (LXA4), resolvins, protectins, and maresins (these lipids and LA, ALA, PGs, LTs, and TXs are termed as bioactive lipids, BALs) [25,26,27,31,32,33]. This assumption is supported by our previous findings that AA, EPA, DHA, LXA4, and resolvin D1 can indeed protect experimental animals from alloxan and STZ-induced type 1 and type 2 DM [16,17,18,34,35,36,37,38,39].

BDNF (brain-derived neurotrophic factor) is known to be produced by several tissues in the body including pancreas, gut, liver, and brain. Previously, it was reported that BDNF is not only a neurotrophic factor but also has anti-diabetic actions [34,40,41,42,43]. In view of its wide distribution and anti-diabetic actions, it is likely that BDNF may serve as a communicator among various tissues especially among brain, liver, pancreas, and gut. Hence, we measured plasma, brain, pancreatic, intestine, and liver tissue content of BDNF in the current study, wherein we investigated the effect of RVD1 against streptozotocin-induced T1DM in male Wistar rats.

## 2. Results

### 2.1. Effect of RVD1 Treatment on Plasma Glucose and Body Weight in STZ Induced T1DM

STZ-induced T1DM was confirmed by measuring plasma glucose levels (>400 mg/dL) after 48 h of induction. It is evident from the results shown in Figure 1B (comparison is between the STZ-induced T1DM group vs. STZ-induced T1DM + RvD1) that RVD1 treatment (*p* < 0.05) reduced the severity of diabetes mellitus by decreasing the plasma glucose levels (Figure 1B) from ~400 to ~400, ~350, ~300, and ~250 mg/dL by the end of 1st, 2nd, 3rd, and 4th week, respectively. Accompanied by this gradual but sustained decrease in blood glucose levels, food consumption was decreased (not shown in the figure), and body weight reached near normal in STZ + RVD1-treated animals (Figure 1C,D respectively). It is seen from the results shown in Figure 1 that RVD1 by itself does not have any effect on basal plasma glucose, insulin, body weight, and food intake.

### 2.2. RVD1 Treatment Decreased Insulin Resistance and Enhanced Insulin Sensitivity

Plasma levels of insulin were lower (*p* < 0.05) in STZ-induced T1DM animals compared to control (Figure 1D) and were restored to normal in the STZ + RVD1 treatment group, suggesting that RVD1 treatment prevents STZ-induced T1DM. STZ-treated animals showed HOMA-IR 3.34 ± 0.5 compared to a control value of 0.66 ± 0.08 (*p* < 0.05) which decreased to 1.28 ± 0.06 (*p* < 0.05) while there was no significant change in RVD1-treated animals compared to the control (0.79 ± 0.04). Similar improvement in the QUICKI index was also seen in STZ + RVD1-treated animals compared to STZ-induced T1DM animals (Control: 0.36 ± 0.001; STZ: 0.21 ± 0.002; RVD1: 0.39 ± 0.003; and STZ + RVD1: 0.36 ± 0.001).

### 2.3. Effect of RVD1 on Tissue BDNF Concentrations and Plasma RVD1 Levels

STZ treatment decreased BDNF concentrations in the brain, pancreas, liver, and intestine tissues (Figure 2A) and were restored to near normal in the STZ + RVD1 group (*p* < 0.01, *p* < 0.001 compared to the control and STZ treatment respectively). STZ-induced T1DM animals showed a significant decrease in plasma RVD1 levels that reverted to near normal in the STZ + RVD1 group (Figure 2B). It is seen from the results shown in Figure 2A,B that RVD1 by itself does not have any effect on the plasma and tissue levels of BDNF and plasma levels of RVD1.

### 2.4. Effect of RVD1 Treatment on STZ-Induced Changes in the Expression of Bcl2/Bax/Pdx/PPAR-γ/Cox-1/Cox-2 Genes

The results of the treatment of RVD1 on the expression of *Bcl2/Bax/Pdx/PPAR-γ/Cox-1/Cox-2* genes is given in Figure 2C. It is seen from these results that RVD1 alone treatment (RVD1 control) did not produce any significant changes in the expression of *Pdx* genes but produced insignificant changes in the expression of *PPAR-gamma*, *Bcl-2*, and *Cox2* (slight increase in the expression) and that of Cox1 and *Bax* (insignificant decrease in the expression). These results suggest that RVD1 does not produce any significant changes in the expression of *Bcl2/Bax/Pdx/PPAR-γ/Cox-1/Cox-2* genes by itself. In contrast to this, as shown in Figure 2C,D, STZ treatment resulted in a significant increase in the expression of *PPAR-gamma*, *Bax*, *Cox1*, and *Cox2* genes in the pancreas (*p* < 0.05) that were restored to near normal in the STZ + RVD1 group. The increase in the expression of *Cox2* seen in STZ-induced T1DM animals (Figure 2C) suggests that STZ has proinflammatory action due to increased formation of prostaglandins (PGs) and leukotrienes (LTs) in T1DM [44,45,46,47]. This increased activity of *COX-2* seen in STZ-treated animals that was restored to normal in the STZ + RVD1-treated group in the present study implies that RVD1 has anti-inflammatory action by virtue of its suppressive action on *Cox2* expression (Figure 2C,D). It is interesting to note that the expression of Pdx was decreased by STZ treatment which was restored to normal in the STZ + RVD1 group. This suggests that RVD1 can enhance the proliferation of pancreatic β cells and thus increase the production of insulin. This may explain as to why RVD1 treatment restored plasma glucose levels to normal in the STZ + RVD1 group compared to STZ-induced T1DM animals (Figure 2C,D).

### 2.5. Effect of RVD1 on Plasma BDNF, Cytokines, and LXA4 Levels in STZ-Induced T1DM

Our previous studies showed that T1DM is associated with decreased β-cell mass due to an increase in pro-inflammatory IL-6 and TNF-α cytokines and a decline in plasma concentrations of LXA4 [16,17,18,36,37,38,39]. The results shown in Figure 3A revealed a significant increase in plasma IL-6 and TNF-α concentrations and a decrease in plasma LXA4 levels in STZ-induced T1DM animals (Figure 3B). IL-6, TNF-α, and LXA4 were restored to near normal levels in the STZ + RVD1-treated group. RVD1 by itself did not have any effect on the plasma levels of IL-6, TNF-α, and LXA4. These results suggest that RVD1 is capable of suppressing plasma pro-inflammatory IL-6 and TNF-α levels that were enhanced by STZ treatment. Similarly, RVD1 treatment (STZ + RVD1 group) restored STZ-induced decrease in plasma LXA4, a potent anti-inflammatory molecule, to near normal levels. There were no significant changes in the plasma levels of LXA4 in the RVD1 alone treated group. These results suggest that RVD1 has potent anti-inflammatory actions.

### 2.6. Effect of RVD1 on STZ-Induced Changes in p65Nf-kB, PPPAR-γ, iNOS, Foxo 1, Gsk3β Proteins in the Pancreatic Tissue

Increase in the expression of pro-inflammatory proteins p65Nf-kB and iNOS (*p* < 0.05) and a decrease in PPAR-γ, Foxo1, and Gsk3β in the pancreatic tissue of STZ-induced T1DM animals was noted (Figure 3C,D). These changes in the proteins p65Nf-kB, PPPAR-γ, iNOS, Foxo 1, Gsk3β reverted to near normal in the STZ + RVD1-treated group.

### 2.7. RVD1 Restores the Balance between Pro- and Antioxidants Altered by STZ

It is evident from the data shown in Table 1 that STZ-induced T1DM animals have significantly (*p* < 0.05) increased plasma nitric oxide (NO) and lipid peroxides (LPO) levels in their pancreatic tissue and reduced (*p* < 0.05) levels of SOD and GST, implying STZ enhances oxidative stress. All these alterations in the antioxidants reverted to near normal in the STZ + RVD1-treated group, suggesting that RVD1 can suppress oxidative stress.

## 3. Discussion

Type1DM that is common in children and young subjects forms 5–10% of the total number of cases of DM. Despite being much less common compared to T2DM, T1DM is important since once the patient is diagnosed to have had this disease, he/she must take lifelong insulin to control hyperglycemia which is a tremendous burden to the patient, patient’s family, and society. Hence, ways of preventing the development of T1DM are urgently needed. Clues as to the involvement of bioactive lipids such as GLA, DGLA, AA, EPA, DHA, LXA4, resolvins, protectins, and maresins in the pathobiology of T1DM are derived from the observation that (i) breast fed children have low incidence of T1DM [28,29,30]; (ii) cod liver oil (rich source of EPA and DHA) supplementation during pregnancy reduces the risk of T1DM in the offspring [31]; (iii) human breast milk is rich in several bioactive lipids [25,26,27,31,32]; (iv) our previous studies which showed that BALs can prevent the development of T1DM in experimental animals [16,17,18,34,35,36,37,38,39]; and plasma concentrations of GLA, DGLA, AA, EPA, DHA, and LXA4 are low chemical-induced T1DM and patients with T1DM [35,36,37,38,39,46,47,48]. These results are in support of the contention that BALs, if not all, some of them may be of significant benefit in the prevention of T1DM. Since all the studies are not possible in humans, some pertinent studies need to be performed in animal models of T1DM. In this context, STZ-induced T1DM is considered as a suitable model for such studies [6].

Results shown in Figure 1 indicate that by the end of the 1st week following STZ + RVD1 treatment, there is no significant decrease in plasma glucose levels. However, a gradual and sustained decrease in plasma glucose levels was noted in STZ +RVD1-treated animals from the end of the 2nd week to the end of the study (4th week), suggesting that there is a gradual regeneration of pancreatic β cells. This is supported by the observation that the expression of the *Pdx* gene (pancreas/duodenum homeobox protein 1) and plasma insulin levels were increased in STZ + RVD1-treated animals in contrast to the suppressive action of STZ on these two indices (Figure 2C,D). Transcription factors are critical to reprogram non-beta cells into insulin-producing β cells, offering a potentially novel regenerative approach for T1DM therapy [27]. Since RVD1 can enhance *Pdx* gene expression, augment insulin secretion, reduce insulin resistance, and enhance insulin sensitivity (see Figure 1 and Figure 2), it is deduced that it (RVD1) is involved in the β cell regeneration process. In a previous study, we noted that LXA4 also enhances *Pdx* expression, prevents STZ-induced type 2 DM, enhances insulin sensitivity, and lowers insulin resistance [17,18], results that are like those seen in the present study. It is noteworthy that STZ-induced suppression of LXA4 secretion is restored to normal by RVD1 (STZ + RVD1 group) treatment (Figure 3B). RVD1 restored to near normal STZ-induced changes in IL-6, TNF-α, and BDNF secretion and altered expressions of *PPAR-γ*, *Bcl-2*, *Bax*, *Pdx*, *COX-1, COX-2* and levels of PPAR, iNOS, Foxo1, and GSK3β proteins (Figure 2 and Figure 3). These results suggest that RVD1 has potent anti-inflammatory actions and regulates down stream of insulin signaling pathways. The results of the present and a previous study [20,49] showed that RVD1 enhances LXA4 secretion which seems to be more potent than RVD1 in its ability to prevent STZ-induced T1DM and T2DM [17,18]. This is so, since LXA4 restored plasma glucose levels in STZ-induced T1DM and T2DM animals by more than 50% from the beginning of the 1st week itself [18]. On the other hand, RVD1 induced a significant decrease in plasma glucose only from the beginning of the 2nd week (see Figure 1B). Since RVD1-treated animals showed a significant increase in plasma LXA4, it is likely that, at least, some of its (RVD1) anti-diabetic and anti-inflammatory actions are because of its ability to enhance LXA4 formation. However, this assumption needs to be confirmed in future studies. RVD1 is known to have a short half-life (from a few seconds to minutes). The enhanced plasma levels of RVD1 observed on day 30 of the present study, despite its administration only for the first 5 days of the experiment, suggests that it (RVD1) may induce its endogenous production in an autocrine fashion.

## 4. Materials and Methods

### 4.1. Chemicals

The reagents and kits used in the present study are given in Table 2. All other fine chemicals and reagents used in the present study were purchased from Sigma Aldrich Chemical Company (St. Louis, MO, USA).

### 4.2. In Vivo Studies

Three to four-week-old Wistar male rats (150–200 gm) purchased from National Institute of Nutrition, (Hyderabad, India) were used for this study and all animal experiments were done following the animal ethics committee and institutional guidelines. The animals were housed at 25 °C room temperature with an alternate 12-h dark and 12-h light cycle. Animals weighing around 190 gm were selected and divided into 4 groups of 6 animals each: controls received PBS and citrate buffer; diabetic group received only STZ; RVD1 group received only RVD1, whereas the treatment group received STZ + RVD1 intraperitoneally.

### 4.3. Induction of Type 1 Diabetes Mellitus

Single intraperitoneal (i.p) injection of freshly prepared STZ (45 mg/kg, dissolved in 50 mM citrate buffer pH 4.5) was given to induce T1DM as described previously [16,17]. The animals developed T1DM in approximately 24 to 48 h after the injection of STZ. Animals that showed significant hyperglycemia and substantial weight loss due to uncontrolled T1DM were used for this study. Freshly prepared RVD1 (60 ng/animal) in 100 µL of 0.1% ethanol in sterile saline) along with or vehicle (100 µL of 0.1% ethanol in sterile saline) was administered for 5 consecutive days via intraperitoneal (i.p) route to the animals from day 1 of STZ injection (Figure 1A). Thus, the first plasma glucose estimation was done on the 7th day of the study which is depicted as the 1st week. Subsequent glucose estimations were done on the 2nd week (day 14), 3rd week (day 21), and 4th week (day 28) and the experiment was terminated on day 30 of the study. Thus, fasting blood glucose levels were measured on day 7, day 14, day 21, and day 28 of the study using the Accu-Check blood glucose meter (Roche, Santa Clara, CA, USA).

### 4.4. Measurement of Blood Glucose, Insulin, Food Intake, and Body Weight

The animals with fasting blood glucose levels >250 mg/dL were confirmed to have developed T1DM. Food consumption and body weight were also measured twice a week. This study was planned for a total duration of 30 days from the day of injection of STZ (Figure 1A). On the 30th day of the study after overnight fasting, plasma glucose and insulin levels were determined. Insulin resistance and sensitivity indices (ISI) were assessed respectively as described previously [17,18,34]. At the end of study, animals were sacrificed, and blood and various tissues (liver, pancreas, intestine, and brain) were collected and stored at −80 °C for further analysis.

### 4.5. BDNF ELISA Studies

Frozen samples of brain, pancreas, liver, and intestine were finely chopped using a sharp scalpel and homogenized in a minimum volume of T-PER (tissue protein extraction lysis buffer with protease inhibitor mixture). Later, the sample was centrifuged for 10,000× *g* for 5 min and the supernatant (with solubilized proteins) was obtained, and the pellet with cell debris was discarded. BDNF Elisa was done in samples of supernatants as per the manufacturer’s instructions and as described previously [34].

### 4.6. Estimation of Antioxidant Enzymes

Lipid peroxides, nitric oxide, along with antioxidant enzymes analysis of super oxide dismutase (SOD), glutathione-S-transferase (GST), glutathione peroxidase (GPX), catalase (CAT) was done in the pancreatic tissue homogenates of all groups of the study as described previously [17,18,19,34].

### 4.7. Measurement of Plasma RVD1, BDNF, TNF-α, IL-6, and LXA4 Concentrations

The plasma levels of RVD1, BDNF (both in the tissues and plasma), IL-6, TNF-α, and LXA4 were measured on the 30th day of the study, which is the end of the study using respective ELISA kits obtained from various manufacturers and in all groups of animals as per the respective manufacturer’s instructions (see Table 1).

### 4.8. Gene Expression Studies in Pancreas

#### 4.8.1. Isolation of RNA and cDNA Synthesis

RNA was isolated from homogenized pancreas using Trizol reagent and cDNAs were synthesized by reverse transcription from 1 μg of total RNA using SuperScript First Strand Synthesis by qRT-PCR (Invitrogen) and according to the manufacturer’s instructions. PCR was performed in Eppendorf 5331 Master Cycler. Quantification of genes was done by Major Science image analysis software.

#### 4.8.2. Semiquantitative PCR for Gene Expression Studies

The cDNAs along with steps and cycles used in gene expression studies were tabulated (Table 3) and PCR products were run on 1.5% (*w/v*) agarose gel in 1× TAE buffer by electrophoresis at 100 V. All experiments were done in triplicate, and statistical analysis was performed based on the ratio of gene of interest transcripts and the amounts of β-actin and calculating as percentage comparing with respective control.

#### 4.8.3. Western Blot Analysis

Primary antibodies including: Nf-kB (P65) (65 kDa), iNOS (130 kDa), Foxo1 (78 kDa), PPAR-γ (53 kDa), Gsk3β (46 kDa), and Actin (45 kDa) along with secondary antibodies were acquired from Cell Signaling Technology, Danvers, MA, USA. Pancreatic tissue is homogenized with T-PER (tissue-protein extraction lysis solution) (50 mg/500 µL lysing solution) along with 10 µL protease inhibitor cocktail and 10 µL phosphatase inhibitor to yield protein lysates. Homogenization was done under ice and centrifuged at 10,000× *g* for 5 min to collect the debris. Quantification of protein was done by Bradford assay and 40 µg of protein was loaded into the SDS-PAGE and later, samples were transferred to the nitrocellulose membrane by using Bio-Rad Transblot turbo transfer system. After transfer, membrane was blocked by using 5% Bio-Rad blocking reagent and incubated with 1:1000 dilution primary rabbit antibodies (Cell Signaling Technology, USA) for 12 h at 4 °C. The membrane was washed after treatment with TBST and blots were incubated with peroxidase conjugated secondary antibody (1: 20,000) treatment for an hour. Using the chemiluminescence document reader, after addition of Immobilon Western chemiluminescent HRP substrate at standardized exposure time, bands were observed. Analysis of samples were done by densitometry analysis compared to β-actin (Major Science image analysis software)

### 4.9. Statistical Analysis

Experiments were performed using six animals per group. All the results are expressed as mean ± SEM. Statistical analysis was done by two-way analysis of variance with Bonferroni post hoc test using Graph pad PRISM analysis software.

## 5. Conclusions

The results of the present study showed that RVD1 has anti-diabetic and anti-inflammatory actions and enhances β cell regeneration. It suppressed insulin resistance and enhanced insulin sensitivity and increased plasma LXA4 and BDNF levels and pancreas, brain, liver, and intestine BDNF concentrations. These results imply that methods designed to augment RVD1 formation may form a new approach to prevent development of T1DM. Based on the results of the present study and previous data [6,13,14,15,16,17,18,34,35,36,37,38,39,40,41,42,43,48,49], the potential mechanism by which STZ could induce T1DM is given in Figure 4.

## Figures and Tables

**Figure 1 ijms-22-01516-f001:**
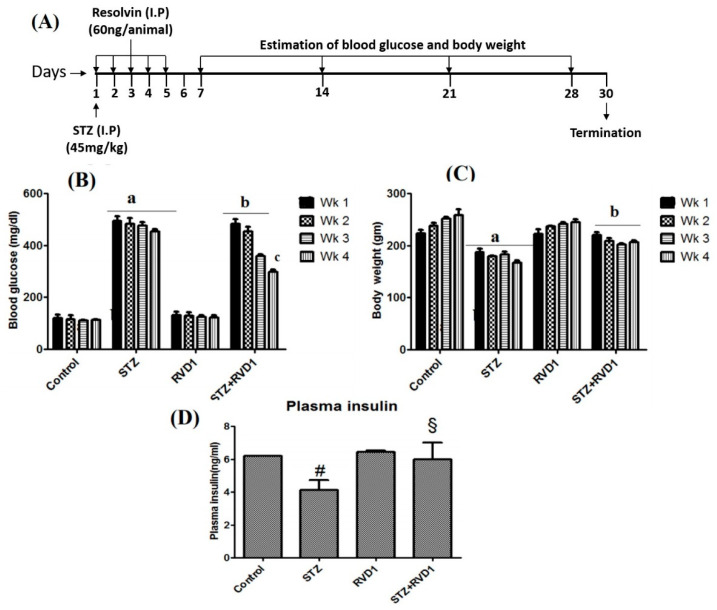
(**A**) Streptozotocin (STZ)-induced type 1 diabetes protocol. Type 1 diabetes mellitus was induced in Wistar rats by single I.P injection of STZ (45 mg/kg) prepared freshly in 50 mM citrated buffer pH 4.5 on (Day 1). On day 1, RVD1 (60 ng/animal intraperitoneally) was injected daily for 5 consecutive days. Blood glucose levels were estimated on day 7, 14, 21, and 28 and animals were sacrificed on day 30. The total duration of this study was 30 days and changes in body weight, food consumption, and blood glucose were monitored weekly. (**B**) Blood glucose levels were measured once on day 7, 14, 21, 28. All values are expressed as mean ± SEM. ^a^
*p* ≤ 0.001 compared to respective weekly control values and ^b^
*p* ≤ 0.01, ^c^
*p* ≤ 0.05 compared to respective STZ–treated group. (**C**) Body weight of rats was done once a week till the end of the study. All the values are expressed as mean ± SEM. ^a^
*p* ≤ 0.01 compared to untreated control and ^b^
*p* ≤ 0.001 compared to STZ. (**D**) Plasma insulin levels were estimated in the plasma collected at the end of the study (day 30); ^#^
*p* ≤ 0.05 compared to untreated control ^§^
*p* ≤ 0.05 compared to STZ. All the above studies were done in each group and consisted of *n* = 6 animals and all values are expressed as mean ± SEM.

**Figure 2 ijms-22-01516-f002:**
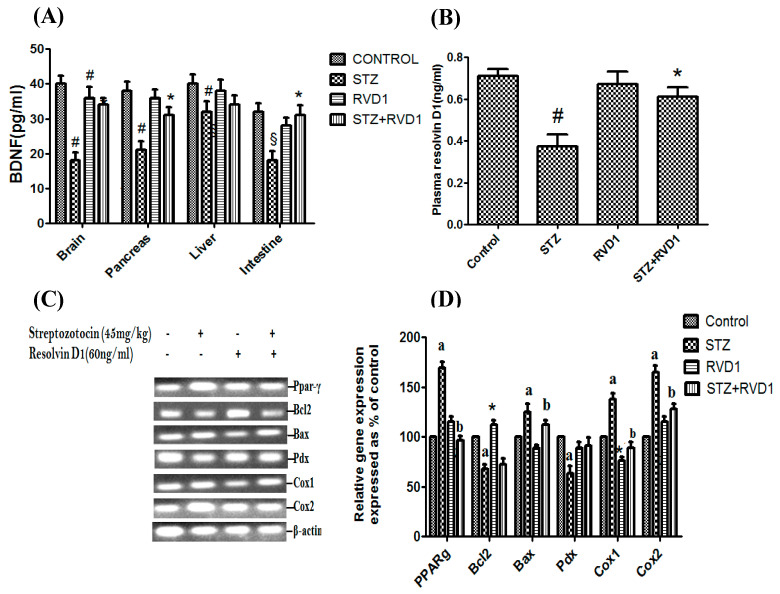
Effect of RVD1 treatment on levels of brain-derived neurotrophic factor (BDNF), plasma RVD1, and gene expression studies by semi-quantitative PCR. (**A**) Relative BDNF protein expression of brain, pancreas, liver, and the intestine of T1DM and control. Data are expressed as mean ± SEM. Brain: control vst1DM: ^#^
*p* ≤ 0.01, * *p* ≤ 0.05; Pancreas: control vs. T1DM: ^§^
*p* ≤ 0.01; Intestine: control vs. T1DM: ^#^
*p* ≤ 0.05. (**B**) Measurement of RVD1 levels in various groups measured at the end of the study (day 30). * *p* ≤ 0.001 compared to untreated control and ^#^
*p* ≤ 0.05 compared to STZ (T1DM) control. (**C**,**D**) Effect of RVD1 (60 ng/kg) treatment on changes in the mRNA expression of *PPAR-gamma*, *Bcl2*; *Bax*; *Pdx*; *Cox1*, and *Cox2* in pancreatic tissue. Pancreatic tissue was collected on day 30. The percentage of change in gene expression and β-actin studied by the semi-quantitative PCR method. The equality of sample loading was confirmed by β-actin gene expression. All the above set of experiments were done on two separate occasions, each time in triplicate (*n* = 6), and all values expressed as mean ± SEM. ^a^
*p* ≤ 0.01, * *p* ≤ 0.05 compared to untreated control values and ^b^
*p* ≤ 0.001 compared to STZ (T1DM) control.

**Figure 3 ijms-22-01516-f003:**
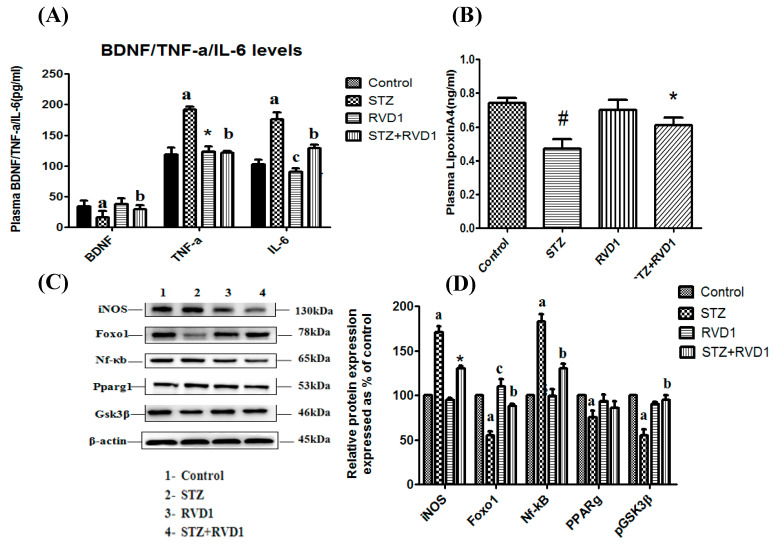
Effect of RVD1 treatment on plasma levels of BDNF/TNF-α/IL-6/LXA4 and protein expression in pancreatic tissue samples. (**A**) Plasma BDNF/TNF-α/IL-6 levels. Plasma BDNF level in STZ + RVD1 vs. STZ (T1DM)-treated groups estimated at the end of the study (day 30). ^b^
*p* ≤ 0.01 compared to STZ (T1DM) and ^a^
*p* ≤ 0.01 compared to untreated control; TNF-α studies: ^a^
*p* ≤ 0.001, * *p* ≤ 0.01 compared to control and compared to STZ control; and ^b^
*p* ≤ 0.05 compared to STZ (T1DM); IL-6 studies: ^a^
*p* ≤ 0.01 and ^c^
*p* ≤ 0.05 compared to untreated control and STZ control values. ^b^
*p* ≤ 0.01 compared to STZ (T1DM) group All values are expressed as mean ± SEM. (**B**) Measurement of LXA4 levels in the plasma of various groups measured at the end of the study (day 30). ^#^
*p* ≤ 0.001 compared to untreated control. * *p* ≤ 0.01 compared to STZ (T1DM) control (positive control group). (**C**,**D**) Protein expression studies in pancreatic tissue of the rats of various groups. Total protein extracted from the pancreatic tissue samples were collected at the end of the study (day 30) and used for Western blots for Nf-κb, Foxo1, PPAR-γ, *p*-GSK3β, iNOS, and beta actin. Equality of loading of the samples was confirmed by beta actin protein expression. All values are expressed as mean ± SEM. ^a^
*p* ≤ 0.01 and ^c^
*p* ≤ 0.05 compared to control values. ^b^
*p* ≤ 0.01 compared to STZ (T1DM).

**Figure 4 ijms-22-01516-f004:**
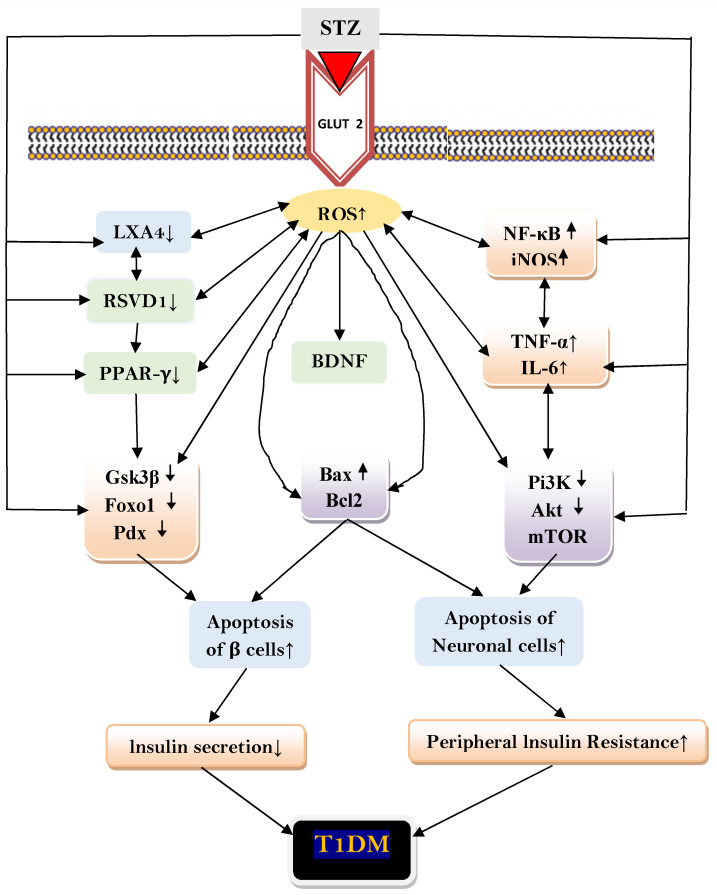
Scheme showing potential mechanism(s) by which STZ could induce T1DM. STZ enters the pancreatic beta cell by binding to GLUT-2 receptors. STZ enhances the generation of ROS (reactive oxygen species) at least in part by activating NADPH oxidase, by enhancing the expression of NF-kB that, in turn, enhances the production of IL-6 and TNF-α. Both these cytokines can also enhance the generation of ROS. Excess ROS generated also includes iNOS (inducible nitric oxide species). ROS and its resultant lipid peroxides can act on Bax and Bcl2, PI3K, Akt and mTOR, PPARγ, Gsk3β, and Pdx and alter their expression in such a way that apoptosis of pancreatic β cells occurs and regeneration of β cells is interfered with leading to decreased insulin production and onset of type 1 DM. STZ either acting directly or through ROS may reduce LXA4 and RSVD1 (also depicted as RVD1 in the text) formation or its half-life that results in their low plasma levels in type 1 DM. Low concentrations of LXA4 and RvD1 and increased Bax and decreased Bcl2 results in apoptosis of pancreatic β cells and development of type 1 DM. LXA4 and RSVD1 have anti-inflammatory actions and thus, decrease peripheral insulin resistance. Decrease in the plasma concentrations of LXA4 and RSVD1 results in an increase in IL-6 and TNF-α levels due to the absence of their (LXA4 and RVSD1) feedback negative control on cytokines. Both IL-6 and TNF-α can also cause increase in peripheral insulin resistance due to their pro-inflammatory actions. The cytoprotective actions of LXA4 and RSVD1 can protect β cells from the cytotoxic action of STZ by suppressing the generation of ROS and inhibiting the formation of IL-6 and TNF-α. RSVD1 enhances LXA4 formation. Thus, there is a close interaction among various lipids. In light of these evidence methods designed to enhance the formation of LXA4/RSVD1, there may form a new therapeutic strategy to prevent type1 DM.

**Table 1 ijms-22-01516-t001:** Concentrations of various anti-oxidants, lipid peroxides, and nitric oxide in pancreatic tissue in vivo: Superoxide dismutase (SOD) is expressed as U SOD/mg of protein; Glutathione-S-transferase (GST) is expressed as µM conjugate formed/minute/gm of protein; Catalase (CAT) is expressed as µM of H_2_O_2_ consumed/minute/mg of protein; Glutathione peroxidase (GPX) is expressed as µg of glutathione consumed/minute/gm of protein. Nitric oxide formed is expressed as µ moles of nitrite formed. Lipid peroxides formed are expressed as µmoles of TMOP formed; * *p* ≤ 0.05, ^§^
*p* ≤ 0.01 vs. untreated control ** *p* ≤ 0.05 compared to STZ group. All values are expressed as mean ± SEM.

Summary of the Analysis of Various Anti-Oxidant/Lipid Peroxides and Nitric Oxide Levels in Pancreas of Resolvin-D1/Stz-Treated Diabetic Rats
Tissue	Group	SOD (Units/mg protein)	CAT (mM H_2_O_2_/min/mg protein	GST (mM/min/gm protein)	GPX (mM/min/gm protein)	LPO (µM TMOP)	Nitric oxide (µM Nitrite)
Pancreas	Control	36.6 ± 1.1	889 ± 69.1	17.7 ± 1.9	116.7 ± 5.7	1.22 ± 0.01	1.06 ± 0.04
STZ (65 mg/kg)	73.1 ± 3.4 *	1291 ± 32 *	51.2 ± 4.5 *	193 ± 3.2 *	1.48 ± 0.21 *	1.22 ± 0.08 *
RVD1	45.3 ± 2.4	732 ± 81.9	27.3 ± 1.5	98.5 ± 2.1 ^§^	0.96 ± 0.10	0.96 ± 0.03 ^§^
STZ + RVD1	41.2 ± 2.9 **	703 ± 60.8 **	21 ± 1.09 **	96.2 ± 5.2 **	0.88 ± 0.06 **	1.08 ± 0.07 **

**Table 2 ijms-22-01516-t002:** Chemicals/kits and reagents along with source used in this in vivo study.

Assay/Reagents	Kit/Analysis	Source
**17(S)-Resolvin D1 (RVD1)**	Analysis	Cayman Chemical Company (Ann Arbor, MI, USA).
**Plasma insulin levels**	ultrasensitive rat insulin ELISA kit	Crystal Chemical Inc (Belvidere, IL, USA)
**Plasma BDNF levels**	BDNF ELISA kit	Chemikine Sandwich ELISA kit (Millipore, Burlington, MA, USA) (CYT306)
**RVD1 levels**	RVD1 ELISA kit	Cayman Chemical Company (Ann Arbor, MI, USA, 500,380)
**IL-6 levels**	IL-6 ELISA kit	Kendall Square (Cambridge, MA, USA)
**TNF-** **α levels**	Quantikine TNF-α Immunoassay ELISA kit	RTA00, R&D Systems (Minneapolis, MN, USA)
**LXA4 levels**	Lipoxin A4 ELISA kit	Oxford Biomedical Research Company (Rochester Hills, MI, USA) (EA45)

**Table 3 ijms-22-01516-t003:** PCR primers (F: forward primer, R: reverse primer) and size of amplicon along with steps in the PCR cycle.

Gene Expression Studies by Semi Quantitative PCR	Size	Primer Sequences	Steps Involved
		Forward/Reverse Primer	Number of Cycles
***Bax***	105 bp	F: 5′-CACCAGCTCTGAACAGATCATGA-3′ R: 5′-TCAGCCCATCTTCTTCCAGATGT-3′	For *Bax/Bcl2* genes, Initial denaturation 94 °C for 2 min, 95 °C for 30 s denaturation, 59 °C for 30 s annealing was employed. Later, 72 °C for 30 s extension and 72 °C for 5 min final extension, and overall, 34 cycles for *Bcl2/Bax.*
***Bcl2***	110 bp	F: 5′-CACCCCTGGCATCTTCTCCTT-3′ R: 5′-AGCGTCTTCAGAGACAGCCAG-3′	
***Pdx***	528 bp	F: 5′-GTAGTAGCGGGACAACGAGC-3′ R: 5′-CAGTTGGGAGCCTGATTCTC-3′	For *β-actin* 94 °C for 2 min initial denaturation, 94 °C for 30 s denaturation, 64 °C and 59 °C for 30 s annealing for *Pdx* and *β-actin*, respectively, and 72 °C for 30 s extension with 72 °C for 5 min final extension and overall, 35 cycles were performed.
***β-actin***	617 bp	F: 5′-CGTGGGCCGCCCTAGGCACCA-3′ R: 5′-TTGGCCTTAGGGTTCAGGGGG-3′	
***PPAR-γ***	859 bp	F: 5′-TGATATCGACCAGCTGAACC-3′ R: 5′-TCAGCGACTGG GACTTTTCT-3′	For *Cox-1*, *Cox-2*, *PPAR-γ* 94 °C for 2 min initial denaturation, 94 °C for 30 s denaturation, 69 °C, 52.5 °C, and for 30 s annealing, respectively. Later, 72 °C for 30 s extension and 72 °C for 5 min final extension, and overall, 35 cycles were performed.
***Cox-1***	450 bp	F: 5′-ACTCACTCAGTTTGAGTCATTC-3′ R: 5′-TTTGATTAGTACTGTAGGGTTAATG-3′	
***Cox-2***	583 bp	F: 5′-TGCATGTGGCTGTGGATGTCATCAA-3′ R: 5′-CACTAAGACAGACCCGTC ATCTCCA-3′	
Polymerase chain reaction (PCR) reagents were obtained from Genei (Bangalore, India). PCR primers were purchased from Bioserve (Hyderabad, India).			

## Data Availability

All the data obtained in this study has been described in full here.

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
