# Peer review of "Resolvin D1 Decreases Severity of Streptozotocin-Induced Type 1 Diabetes Mellitus by Enhancing BDNF Levels, Reducing Oxidative Stress, and Suppressing Inflammation"

_ijms, 2021, doi:10.3390/ijms22041516_

Round 1
Reviewer 1 Report
Here the authors use a STZ rat model to identify if Resolvin D1 reduces inflammation and subsequently diabetes. To do this they have used STZ injection alongside RVD1 administration. RVD1 administration restored levels of proinflammatory cytokines and apoptotic factors. Although an interesting relevant study about the DHA derivative Resolvin D1, it is this reviewers opinion that some limitations exist in the current form:
- Overall more clarity is required in the manuscript. Very limited descriptions in the introduction and results section. The manuscript is about Resolvin D1 yet the authors spend only 4 lines in the introduction on this topic. Please expand. Why choose the STZ rat model? It is stated in the beginning of the discussion, it would be better placed in the introduction.
- It is this reviewers opinion that the results section and figures would be better separated into the effects of RVD1 without induction of T1D – and then RVD1 +STZ. As clearly RVD1 induces effects on the pancreas without diabetes being induced? This would clarify results. In methods it is stated rats received vehicle control for RVD1 but no vehicle group is shown? This should be shown on the graphs or at least reported.
- Histology should be shown to highlight changes in beta cell mass in the pancreas or fewer destroyed pancreatic islets.
Minor points:
- Sentence in the introduction from line 43 to 47 should be edited.
- Figure 1a legend should state what ‘C’ is.
- Figure 1b, bar for statistics is unclear what is being compared.
- No stats on Figure 1c. Yet, statistical information in the legend. Please clarify.
- What is the rationale of studying the BDNF levels in brain, liver and intestine in this study? Please clarify why these tissues were chosen.
- Explain in the results sections, where statistical comparisons are made with which group.
- No mention of pdx gene in results 3.4 section but in discussion.
- Line 264 states that there is an increase in LXA4 – is this the case in 3B? no difference to control? Please clarify lines 264-265. Is it the case that LXA4 is only increased by RVD1 under STZ conditions?
- Please state how many experiments these data represent. Do these experiments represent one experiment only?
- Line 68 should not have a full stop after STZ+RVD1
- The effects of RVD1 may be better presented in a diagram.
Author Response
//
Reviewer # 1:
Comment # 1: Here the authors use a STZ rat model to identify if Resolvin D1 reduces inflammation and subsequently diabetes. To do this they have used STZ injection alongside RVD1 administration. RVD1 administration restored levels of proinflammatory cytokines and apoptotic factors. Although an interesting relevant study about the DHA derivative Resolvin D1, it is this reviewers opinion that some limitations exist in the current form:
Overall, more clarity is required in the manuscript. Very limited descriptions in the introduction and results section. The manuscript is about Resolvin D1 yet the authors spend only 4 lines in the introduction on this topic. Please expand. Why choose the STZ rat model? It is stated in the beginning of the discussion, it would be better placed in the introduction.
Response: As suggested by the reviewer, we have expanded both Introduction and Discussions sections. Added more information about RVD1 and the rationale of the study and why we selected STZ-induced model of T1DM.
Comment # 2: It is this reviewers opinion that the results section and figures would be better separated into the effects of RVD1 without induction of T1D – and then RVD1 +STZ. As clearly RVD1 induces effects on the pancreas without diabetes being induced? This would clarify results. In methods it is stated rats received vehicle control for RVD1 but no vehicle group is shown? This should be shown on the graphs or at least reported.
Response: As suggested by the reviewer in the results section, we have added more information about the effects of RVD1 alone on various indices studied and identified STZ + RVD1 treatment actions separately. In each group there were 6 animals, and this has been mentioned in the Methods section and highlighted in red for easy identification. We have not done histopathology in the present study. But we have done his previously and showed that in the group that received the lipids along with STZ/alloxan have more beta cells.
Comment # 3: Figure 1b, bar for statistics is unclear what is being compared.
Response: As suggested by the reviewer, the comparisons are as follows: §P≤0.001 and ₰P≤0.01 compared to respective weekly control values and **P≤0.01, *P≤0.05 compared to respective STZ–treated group. This has been clarified in the revised version under Legends to Figures and tables.
Comment # 4: No stats on Figure 1c. Yet, statistical information in the legend. Please clarify.
Response: This error has bene corrected.
Comment # 5: What is the rationale of studying the BDNF levels in brain, liver and intestine in this study? Please clarify why these tissues were chosen.
Response: The reason as to why we studied BDNF levels has been discussed in the revised Introduction section.
Comment # 6: Explain in the results sections, where statistical comparisons are made with which group.
Response: As suggested by the reviewer, this has been done in the revised version of the manuscript.
Comment # 7: No mention of pdx gene in results 3.4 section but in discussion.
Response: As suggested by the reviewer, a brief mention and discussion about Pdx expression has been inserted in the revised Results section of the manuscript.
Comment # 8: Line 264 states that there is an increase in LXA4 – is this the case in 3B? no difference to control? Please clarify lines 264-265. Is it the case that LXA4 is only increased by RVD1 under STZ conditions?
– is this the case in 3B?
Response: Yes. There is no significant change between control and RVD1 treatments. LXA4 was increased (to near normal values) only in the STZ + RVD1 group.
Comment #9: Please state how many experiments these data represent. Do these experiments represent one experiment only?
Response: These data represent one experiment wherein 6 animals were used per group.
Comment # 10: Line 68 should not have a full stop after STZ+RVD1.
Response: Done as suggested. We apologize for the error.
Comment # 11: The effects of RVD1 may be better presented in a diagram.
Response: As suggested by the reviewer, we have added Figure 4 in the revised manuscript.
Reviewer 2 Report
Statistics – Should be rerun using Two-way Analysis of Variance with the appropriate post hoc test. For analyses that involved repeated measures such as for body weight and blood glucose a repeated measures analysis should be used.
Figure 1. Please re-graph the data in this figure for C & D. To utilize a line graph the points to be connected by a line must be related to one another. Thus, if the x axis is presented as time in weeks and the symbols/lines represented different groups then a line graph could be used. As presented, it is an incorrect use of the format.
Also, the anomalous impact of RVD1 alone treatment on blood glucose concentration shown in Figure 1B should be further discussed. The mean values do not seem to differ from the STZ treatment yet no negative impacts appear to be manifested.
Section 3.2 – The text on lines 162 to 165 is not so clear to this reviewer as to which treatment is the combination of STZ-induced plus RVD1 versus RVD1 alone.
Figure 2 legend – D is not identified in the legend. Presumably the “D)” is missing in front of the sentence that starts ..”The percentage of change..” on line 182.
Lines 189-190. The way that sentence reads does not agree with the graph and indicated statistics in Figure 2C. To this reader the STZ treatment led to the greatest value for PPAR-gamma versus control rather than RVD1 treatment. Moreover, STZ treatment upregulated rather than suppressed PPAR-gamma as per the figure. The BCL-2 response seems to reflect the description in this sentence rather than PPAR-gamma response.
Discussion lines 268 – 270. This needs to be tested directly – just speculation. Might indicate how this idea that RVD1 actions mediated by LXA4 can be tested.
Author Response
Comment # 1: Statistics – Should be rerun using Two-way Analysis of Variance with the appropriate post hoc test. For analyses that involved repeated measures such as for body weight and blood glucose a repeated measures analysis should be used.
Response: As suggested by the reviewer, we have analyzed the data employing two-way Anova – Bonferroni posttests tool in Graph pad PRISM analysis software.
Comment # 2: Figure 1. Please re-graph the data in this figure for C & D. To utilize a line graph the points to be connected by a line must be related to one another. Thus, if the x axis is presented as time in weeks and the symbols/lines represented different groups then a line graph could be used. As presented, it is an incorrect use of the format.
Response: As suggested by the reviewer, the errors have bene corrected in the revised manuscript.
Comment # 3: Also, the anomalous impact of RVD1 alone treatment on blood glucose concentration shown in Figure 1B should be further discussed. The mean values do not seem to differ from the STZ treatment, yet no negative impacts appear to be manifested.
Response: As suggested by the reviewer, we have discussed the effect of RVD1 treatment on blood glucose levels under Results section.
Comment # 4: Section 3.2 – The text on lines 162 to 165 is not so clear to this reviewer as to which treatment is the combination of STZ-induced plus RVD1 versus RVD1 alone.
Response: The lines have been corrected as suggested by the reviewer.
Comment # 5: Figure 2 legend – D is not identified in the legend. Presumably the “D)” is missing in front of the sentence that starts ..”The percentage of change..” on line 182.
Response: As suggested by the reviewer, this error has been corrected. It may be noted that under legends, Fig 2 C-D were clubbed together, and description was given.
Comment # 6: Lines 189-190. The way that sentence reads does not agree with the graph and indicated statistics in Figure 2C. To this reader the STZ treatment led to the greatest value for PPAR-gamma versus control rather than RVD1 treatment. Moreover, STZ treatment upregulated rather than suppressed PPAR-gamma as per the figure. The BCL-2 response seems to reflect the description in this sentence rather than PPAR-gamma response.
Response: This error has been corrected in the revised version of the manuscript.
Comment # 7: Discussion lines 268 – 270. This needs to be tested directly – just speculation. Might indicate how this idea that RVD1 actions mediated by LXA4 can be tested.
Response: We agree with the reviewer that this is a speculation based on the results obtained and results of a previous study (see reference 49) that showed that RVD1 treatment can increase LXA4 production. IN view of this, we have added a sentence in the discussion section that this needs to be verified in future studies.

Round 2
Reviewer 1 Report
Clarity much improved with the additional text in the introduction and Figure 4.
Minor comments:
Line 220 should say 'Fig. 1' not Fig. 2
Line 238 there is no 'Fig. 1E'.
Figure 2 - the authors state no significant differences between RVD1 alone and control on expression of bcl2 and cox1 - but on figure 2D * p<0.05 compared to 'control' - can the authors clarify?
Figure 3A, the legend in the figure is now incorrect.
Figure 3A, TNFa - *p<0.05 compared to control. IL-6 'c' p<0.05 compared to control. Is this correct? Do the authors mean compared to STZ?
Figure 3 legend - 'D' needs to be included
Figure 4 legend is incorrect. Figure 4 legend would be better expanded.
Author Response
/
I am here with submitting my revised manuscript: Manuscript ID: ijms-1022990
Title: Resolvin D1 decreases severity of streptozotocin-induced type 1
diabetes mellitus by enhancing BDNF levels, reducing oxidative stress and
suppressing inflammation. In this modified version, all the suggestions made by the editorial office/reviewers have been addressed. I wish to add the following in response to specific comments made by the reviewers.
Minor comments:
Comment: Line 220 should say 'Fig. 1' not Fig. 2.
Response: Error has been corrected.
Comment: Line 238 there is no 'Fig. 1E'.
Response: Error has been corrected.
Comment: Figure 2 - the authors state no significant differences between RVD1 alone and control on expression of bcl2 and cox1 - but on figure 2D * p<0.05 compared to 'control' - can the authors clarify?
Response: This is in comparison to untreated control-error has been corrected.
Comment: Figure 3A, the legend in the figure is now incorrect.
Response: The legend of Fig 3A is correct.
Comment: Figure 3A, TNFa - *p<0.05 compared to control. IL-6 'c' p<0.05 compared to control. Is this correct? Do the authors mean compared to STZ?
Response: Error has been corrected.
Comment: Figure 3 legend - 'D' needs to be included
Response: “D’ legend included along with “C”.
Comment: Figure 4 legend is incorrect. Figure 4 legend would be better expanded.
Response: Legend expanded as suggested.
I trust that the revised manuscript is now in an acceptable form since we have answered all the comments of the reviewers.
I wish to thank the reviewers for their incisive and excellent comments that made the manuscript better.
All the corrections are highlighted in red.
If any further clarifications are needed, kindly inform us.
I trust that the manuscript is now in an acceptable form for early publication. If you need any further clarifications, kindly let me know.
